# Flavonoids from *Hippophae rhamnoides* Linn. Revert Doxorubicin-Induced Cardiotoxicity through Inhibition of Mitochondrial Dysfunction in H9c2 Cardiomyoblasts In Vitro

**DOI:** 10.3390/ijms24043174

**Published:** 2023-02-06

**Authors:** Wenna Zhou, Jian Ouyang, Na Hu, Honglun Wang

**Affiliations:** 1Department of Pharmaceutical Engineering, School of Life and Health Sciences, Huzhou College, Huzhou 313000, China; 2Qinghai Provincial Key Laboratory of Tibetan Medicine Research and CAS Key Laboratory of Tibetan Medicine Research, Northwest Institute of Plateau Biology, Xining 810008, China; 3Huzhou Plateau Biological Resource Centre of Innovation, Northwest Institute of Plateau Biology Chinese Academy of Sciences, Huzhou 313000, China; 4University of Chinese Academy of Sciences, Beijing 100049, China

**Keywords:** doxorubicin, flavonoids, *Hippophae rhamnoides* Linn., JNK-Sab-Ros

## Abstract

Doxorubicin (Dox) is one of the most frequently prescribed anti-cancer drugs. However, treatment with Dox is limited due to cumulative cardiotoxicity. 3-*O*-β-d-Sophorosylkaempferol-7-*O*-{3-*O*-[2(E)-2,6-dimethyl-6-hydroxyocta-2,7-dienoyl]}-α-L-rhamnoside (F-A), kaempferol 3-sophoroside 7-rhamnoside (F-B), and hippophanone (F-C) were successfully obtained by purification and separation of seabuckthorn seed residue in our previous research. This study was undertaken to investigate the protective effect of three flavonoids against Dox-induced H9c2 cell apoptosis. Cell proliferation was detected by MTT assay. 2′,7′-Dichlorofluorescein diacetate (DCFH-DA) was used to determine the production of intracellular reactive oxygen species (ROS). ATP content was measured using an assay kit. Transmission electron microscopy (TEM) was used to observe changes in mitochondrial ultrastructure. The expression levels of proteins (p-JNK, JNK, p-Akt, Akt, p-P38, P38, p-ERK, ERK, p-Src, Src, Sab, IRE1α, Mfn1, Mfn2, and cleaved caspase-3) were evaluated by Western blot. Molecular docking was performed using AutoDock Vina. The three flavonoids could significantly relieve Dox-induced cardiac injury and inhibit cardiomyocyte apoptosis. The mechanisms were mainly related to the stability of mitochondrial structure and function maintained by suppressing the production of intracellular ROS, p-JNK and cleaved caspase-3, and increasing ATP contents and protein expression of mitochondrial mitofusin (Mfn1, Mfn2), Sab and p-Src. Pretreatment with flavonoids from *Hippophae rhamnoides* Linn. can reduce Dox-induced H9c2 cell apoptosis based on the ‘JNK-Sab-Ros’ signal pathway.

## 1. Introduction

Doxorubicin (Dox), as an anthracycline anti-cancer drug, is widely used in chemotherapy [1,2]. It has efficacy in fighting several solid tumors, including in breast cancer, lung cancer, and leukemia [3,4]. As with other anticancer agents, the utility of Dox in the clinic is limited by the risk of toxicity. The previous research results show that the heart is a preferential target of Dox toxicity. Cumulative, dose-related, progressive myocardial damage may further lead to congestive heart failure [5]. Thus far, it is not very clear whether cellular and molecular pathogenesis is associated with Dox-induced heart failure. 

A possible molecular mechanism is widely accepted: doxorubicin can induce and promote reactive oxygen species (ROS) generation, in turn leading to DNA damage or mitochondrial dysfunction. Doxorubicin-induced mitochondrial damage leads to the release of inner membrane proteins that trigger the activation of the caspase family, such as caspase-3, then furthering induce cell apoptosis [6,7]. The production of free radicals is a major mechanism through which Dox injures the myocardium [8,9,10]. In addition, more than 90% of the ATP utilized by cardiomyocytes are produced by mitochondria [11]. Dox-induced cardiotoxicity can lead to significant mitochondrial structural dysfunction [12,13]. One of the mechanisms suggested is that mitochondria play an important role in Dox-induced intrinsic apoptosis [14]. Myocardial cell death as the terminal downstream event is triggered by free radical production and mitochondrial dysfunction. Dox cardiotoxicity emphasizes how we urgently need new treatment protocols to protect the heart without diminishing the drug’s anti-tumor activity. 

Total flavonoids of seabuckthorn are used as pharmaceutical ingredients to treat cardiovascular diseases. Tongxinshu capsules, mainly used in patients with coronary heart disease and angina pectoris, are manufactured by Qinghai high tech Pharmaceutical Co., Ltd. and include seabuckthorn flavonoids as a main ingredient [15]. However, the mechanisms underlying the cardioprotective effects of seabuckthorn flavonoids remain undefined. Abudureyimu Gulimire et al. reported that total flavonoids from seabuckthorn can increase myocardial GSH-Px and SOD activities and decrease MDA content to protect myocardial function against Dox-induced cardiotoxicity [16]. Another result indicated that isorhamnetin, the most abundant flavonol in seabuckthorn, can counteract Dox-induced oxidative stress and suppress the activation of the mitochondrion apoptotic pathway and mitogen-activated protein kinase pathway [17,18]. 3-*O*-β-d-Sophorosylkaempferol-7-*O*-{3-*O*-[2(E)-2,6-dimethyl-6-hydroxyocta-2,7-dienoyl]}-α-L-rhamnoside (F-A), kaempferol 3-sophoroside 7-rhamnoside (F-B), and hippophanone (F-C) were successfully separated and purified by our laboratory, and their chemical structures are shown in Figure 1 [19]. In addition, previous studies have also found that the alkaloids or phenylpropanoids in seabuckthorn have the effect of alleviating myocardial toxicity [20,21,22]. The purpose of our research is to test the hypothesis that the flavonoids from seabuckthorn could prevent Dox-induced cardiotoxicity and to identify the involved initial mechanisms in an in vitro model. 

## 2. Results

### 2.1. Flavonoids from Hippophae rhamnoides Linn. Inhibits Dox-Induced Cytotoxicity in H9c2 Cells

H9c2 cells were pre-treated for 1 h with F-A, F-B, and F-C (0.1, 1.0, 5.0, 10, 20, 40, 80, and 160 μM) in the presence or absence of 2.5 µM Dox (shown in our previous studies) [19]. As is shown in Figure 2A–C, no significant toxic effect was observed in the following treatment with F-A, F-B, and F-C without Dox. F-A, F-B, and F-C compared to the Dox model group all significantly decreased Dox-induced cytotoxicity as shown in Figure 2D–F. The amount of 10 µM was chosen for this study in consideration of raw material and cell viability. 

### 2.2. Flavonoids Significantly Decrease Dox-Induced Cell Apoptosis

In order to investigate the Dox-induced cell apoptosis, the expression level of cleaved caspase-3 were examined (Figure 3A,B). The relative expression level of cleaved caspase-3 significantly increased compared to the normal control group (NC group) (*p* < 0.01). There was no significant activation of cleaved caspase-3 in the F-A, F-B, and F-C group. However, F-A, F-B, and F-C (10 μM) significantly inhibited the activation of caspase-3 in different degrees in comparison to the Dox model group (*p* < 0.01).

### 2.3. Flavonoids Inhibit JNK Activation in Dox-Treated H9c2 Cells

H9c2 cells treated with 2.5 μM Dox rapidly induced the relative expression level of phosphorylation of JNK (p-JNK, 2.38 folds vs. control group, *p* < 0.01), as shown in Figure 3C,D. No significant increase compared to the NC group (*p* > 0.05) in the expression level of p-JNK was shown in the following treatment with 10 μM F-A, F-B, and F-C with the absence of Dox. The expression levels of p-JNK compared to the Dox group was significantly suppressed by the administration of F-A, F-B, and F-C under the 10 μM concentration condition in the presence of 2.5 µM Dox. However, F-A and F-C pretreatments were lower effective than that of F-B at the 10 µM dosage in preventing the increase of p-JNK. As is shown in Figure 4, the expression levels of p-P38, p-ERK, and p-Akt were sharply increased compared to the NC group by 2.5 μM Dox. However, F-A, F-B, and F-C had no protective effect to reduce protein phosphorylation. Therefore, we are going to focus on JNK in further research.

### 2.4. Flavonoids Effectively Improved Dox-Induced Mitochondrial Dysfunction in H9c2 Cells

Flavonoids inhibit Dox-induced increases in the levels of intracellular ROS in H9c2 cells. Compared to the NC group, 2.5 µM Dox added to H9c2 cells caused a ~2.5-fold increase in the DCF fluorescence intensity; conversely, the F-A, F-B, and F-C groups showed no significant increase (*p* > 0.05), as shown in Figure 5A. Pre-treatment with 10 μM F-A, F-B, or F-C can inhibitDox-induced elevation of intracellular ROS level (*p* < 0.01). The protective effects of F-B and F-C on intracellular ROS level were better than those of F-A. 

Flavonoids raise the Dox-induced reduction of intracellular ATP contents. Our data suggested that Dox treatment caused a decrease in intracellular ATP synthesis as shown in Figure 5B. Compared to the NC group, there is no a significant reduction in the relative content of ATP in F-A, F-B, or F-C group (*p* > 0.05). However, pre-treatment with 10 µM F-A, F-B, or F-C could significantly raise the Dox-induced reduction of intracellular ATP relative contents compared to the Dox group (F-A vs. Dox group, *p* < 0.05; F-B vs. Dox group, *p* < 0.01; F-C vs. Dox group, *p* < 0.01). The results show that F-B prevents the decrease in intracellular ATP relative contents better than F-A or F-C.

To confirm the results of the mitochondrial biogenesis assessment, immunoblot analysis for key mitochondrial fusion proteins (Mfn1, Mfn2) was conducted. Dox treatment significantly decreased the protein level of Mfn1 and Mfn2 (*p* < 0.01) in comparison to the NC group as shown in Figure 6A,C. However, pretreatment with the F-A, F-B, or F-C group in the absence of 2.5 µM Dox did not significantly decrease the protein level of Mfn1 and Mfn2 in comparison to the NC group (*p* > 0.05). Pretreatment with F-A, F-B, and F-C followed by 2.5 μM Dox significantly increased the protein level of Mfn1 and Mfn2 in comparison to the Dox group (*p* < 0.01, Figure 6B; *p* < 0.01, Figure 6D).

Electron microscopy was used to gain insight into structural changes caused by exposure to 2.5 μM Dox for 24 h. Cells treated with the 2.5 μM Dox group showed mitochondrial alterations, including enlargement, matrix disorganization, and loss of mitochondrial cristae in comparison to the NC group (Figure 7B). Cytosolic vacuoles become apparent at this concentration. Three flavonoids of seabuckthorn pre-treatment effectively reduced Dox-induced mitochondrial structural abnormalities in H9c2 cells, as shown in Figure 7F–H.

### 2.5. Flavonoids Based on the “JNK-Sab-ROS” Pathway Effectively Alleviate Dox-Induced Cardiotoxicity

Dox treatment significantly decreased the protein level of Sab and p-Src (*p* < 0.01) in comparison to the NC group as shown in Figure 8A. However, pretreatment with the F-A, F-B, or F-C group in the absence of 2.5 µM Dox did not significantly decrease the protein level of Sab and p-Src in comparison to the NC group (*p* > 0.05). Dox treatment significantly increased the protein level of p-IRE1α (*p* < 0.01) in comparison to the NC group as shown in Figure 8A,B. Pretreatment with F-A and F-C followed by 2.5 μM Dox significantly increased the protein level of Sab in comparison to the Dox group (*p* < 0.05, Figure 8B). Pretreatment with F-B and F-C followed by 2.5 μM Dox significantly increased the protein level of p-Src in comparison to the Dox group (*p* < 0.05, Figure 8D). 

### 2.6. Molecular Docking of JNK1 and Three Flavonoids

From the results, F-A interacted with Gly-35, Ala-36, Arg-69, Asn-114, Ser-155, Asn-156, Leu-168, Gly-171, Thr-188, and Arg-192 residues in JNK1 via van der Waals forces, as shown in Figure 9A. For the interaction between F-B and JNK1 in Figure 9B, a hydrogen bond was suggested at Gly-35, Ala-36, Arg-69, Asn-114, Ser-155, Asn-156, Thr-188, and Arg-192 residues. Similarly, F-C interacted with Lys-55, Arg-69, Met-111, Asn-114, Ser-155, Val-186, Thr-188, and Arg-192 residues in JNK1, as shown in Figure 9C. Ser-155, Thr-188, and Arg-192 are three key sites. Additionally, the binding energies between F-A, F-B, and F-C with JNK1 were found to be −10.9 kcal/mol, −10.4 kcal/mol, and −11.5 kcal/mol, respectively. 

## 3. Discussion

Although Dox is an effective anti-cancer chemotherapeutic drug, its clinical application is limited by cumulative, dose-related, progressive myocardial damage, which may further lead to severe and irreversible cardiomyopathy and heart failure [7,23]. Flavonoids have been described as inhibitors of Dox-induced H9c2 cell apoptosis in several other reports [24]. There is no research on the pharmacological activity of F-A, F-B, and F-C flavonoids. However, a study has shown that total flavonoids of seabuckthorn can protect the heart against Dox-induced cardiotoxicity, a mechanism which is associated with antioxidant activity [16]. Moreover, seabuckthorn flavonoid extracts are used as pharmaceutical ingredients to treat cardiovascular diseases in many traditional Chinese patent medicines, such as Tongxinshu capsules. Our previous in vitro study showed that the antioxidant activity of these three flavonoids was excellent [25]. The study hypothesized that seabuckthorn flavonoids have a protective effect on Dox-induced cardiotoxicity in H9c2 cells. Firstly, relative cell activity results indicate that F-A, F-B, and F-C all protected H9c2 cells from Dox-induced cytotoxicity. How can they play a protective role in Dox-induced cardiotoxicity?

A mechanism that has gained general acceptance is that ROS, as a critical initiator, triggers myocardial cell apoptosis. Several existing reports have shown that the mitochondrial electron transport chain (ETC) is the major source for ROS generation in cardiac cells [26]. The semiquinone form of Dox is considered as a toxic short-lived metabolite, which can interact with mitochondrial enzymes and elicit the accumulation of a lot of ROS [7]. The cardiac tissue is richer in mitochondria than in other tissues. Mitochondria make up about 45% of the myocardial volume [27]. The heart is particularly susceptible to oxidative stress because of its high energy demand that is satisfied by mitochondrial respiration, which generates ATP. Therefore, the greater the number of mitochondria in an organ, the stronger the interaction with the semiquinone form of Dox, and the more ROS are produced. Mitochondria are injured by Dox-induced cumulative ROS, which results in the impaired expression of ETC complex proteins, further leading to more ROS generation. Then, a vicious circle is formed that exacerbates cell apoptosis. The activation of caspase-3 is a vital step in Dox-induced apoptosis [23]. 

Our results showed that the protective effect of F-A, F-B, and F-C was mediated through the decrease of ROS levels and increase of ATP content as shown in Figure 3. We further explored whether F-A, F-B, and F-C inhibit apoptosis-related cleaved caspase-3 protein expression level. They did inhibit apoptosis as shown in Figure 4A. These results suggest that they can ameliorate Dox-induced H9c2 cell damage. What is the possible molecular mechanism between them? We hypothesized that this protective effect may be related to the MAPK pathway and mitochondrial structural integrity. Then, we further explored and investigated the protein expression levels of p-JNK, p-ERK, p-P38, p-Akt, Mfn1, and Mfn2. Dox-induced mitochondrial fragmentation is associated with the upregulation of dynamin-related protein 1 (Drp1) and the downregulation of fusion proteins (Mfn1, Mfn2), which leads to cytochrome c release and the activation of caspases [28]. In this study, compared to the NC group, the Dox-treated group was able to significantly inhibit the protein expression level of Mfn1 and Mfn2. Pretreatment with 10 μM F-A, F-B, and F-C can improve this trend that was induced by Dox (Figure 5). ROS acts as a stimulus and modulates mitogenic-activated protein kinase (MAPK) pathways, which comprise extracellular-regulated (ERKs), c-jun-NH2-terminal kinase (JNKs), and p38 MAPK [29,30]. Several studies’ results indicate that persistent phosphorylation of JNK and ERK1/2 can be proapoptotic in Dox-treated cardiac muscle cells [31]. In our study, F-A, F-B, and F-C-treated H9c2 cells significantly inhibited the Dox-induced activations of JNK. However, there was no significant reduction the in the levels of p-ERK1/2 and p-P38 in comparison with the Dox group (Figure 4). We speculate that the three flavonoids’ reduction of Dox-induced apoptosis may be achieved through the downregulation of ROS-dependent JNK activation that further promotes cell survival, rather than through the downregulation of ERK, P38, and p-Akt activation. 

Sab, a c-Jun N-terminal kinase (JNK)-interacting protein, is located at the mitochondrial outer membrane, which mediates the mutual amplification of mitochondrial ROS generation and JNK activation [32]. That is the reason we focused on the mitochondrial JNK–Sab–ROS positive feedback loop to elucidate the mechanism of Dox-induced apoptosis. Western blotting results indicated that F-A, F-B, and F-C all protected H9c2 cells from Dox-induced cytotoxicity. Pretreatment with F-A and F-C significantly increased the protein level of Sab in comparison to the Dox group. However, pretreatment with F-B and F-C increased the protein level of p-Src. This inconsistency of results may be due to the regulatory effects of other signaling pathways. 

To verify whether the three flavonoids exert a protective effect on Dox-treated H9C2 cells mainly by inhibiting the phosphorylation of JNK, molecular docking between the flavonoids and JNK1 was performed. The optimal binding conformation of the complex is shown in Figure 9. The results suggest that the binding between the three flavonoids and JNK1 is stable and forms a favorable conformation, in agreement with their protective effects in Dox-induced H9c2 cells. Moreover, the results potentially indicated that flavonoids can exert a cardioprotective effect by inhibiting the phosphorylation of JNK. How does JNK play a key role in Dox-induced cardiotoxicity in mitochondrial damage and reactive oxygen species levels? 

These results demonstrated that flavonoids from *Hippophae rhamnoides* Linn. revert doxorubicin-induced cardiotoxicity through inhibition of mitochondrial dysfunction in H9c2 cardiomyoblasts in vitro.Further studies showed that the mechanism was mainly related to uppressing the production of intracellular ROS, reducing activation of JNK, and decreasing expression level of cleaved caspase-3 protein, increasing ATP contents, favoring mitochondrial mitofusin (Mfn1, Mfn2) and regulating Sab and p-Src protein expression. Flavonoids from Hippophae rhamnoides Linn. may be based on the “JNK–Sab–ROS” pathway to revert Dox-induced cardiotoxicity; the possible signaling mechanism pathways are shown in Figure 10. F-A, F-B, and F-C may be attractive candidates as Dox-induced cytotoxicity improving agents in clinical treatment to protect against cardiotoxicity in Dox-exposed patients in the future. However, more investigations are needed to elucidate the probable underlying mechanisms of these valuable effects.

## 4. Materials and Methods

### 4.1. Preparation of Purified Compounds

Three flavonoids from seed residue of *Hippophae rhamnoides* L. were isolated and purified by our laboratory as described in previous research [19]. The isolated flavonoids were prepared in dry powder and stored at −20 °C.

### 4.2. Cell Culture

H9c2 embryonic rat cardiac cells (H9c2 cells) were purchased from the cell bank of the Institute of Biochemistry and Cell Biology of Shanghai (Shanghai, China) and were cultured in Dulbecco’s modified Eagle’s medium (DMEM; Gibco, Carlsbad, CA, USA) containing 10% fetal bovine serum (FBS; Gibco, Carlsbad, CA, USA) and 1% penicillin streptomycin, 100 μg/mL of streptomycin (Gibco, Grand Island, NY, USA), and 5% CO_2_ at 37 °C. The cells were fed every 2~3 days and subcultured once they reached 80% confluence. Cells in the logarithmic growth phase were used for further experiments.

### 4.3. Cell Viability Assay

The cell viability was determined using a modified MTT assay as described previously [33]. H9c2 cells (7500/well) were seeded out in 96-well culture plates in 200 μL volumes. Different concentrations of seabuckthorn flavonoids were pretreated 1 h before 2.5 μM Dox treatment and then co-incubation was continued with 2.5 µM doxorubicin for 24 h. The 3-(4,5-dimethylthiazol-2yl)-2,5-diphenyltetrazoliumbromide (MTT) solution (5 mg/mL in PBS) was added to each well. MTT was purchased from Sigma Aldrich Inc. (St. Louis, MO, USA). Then, 96-well plates were incubated for 3 h in the incubator. The resultant formazan product was dissolved by DMSO (100 μL). Absorbance was detected at 570 nm using a Multi-Mode Detection Platform (Molecular Devices, San Jose, CA, USA). The experiment was carried out in triplicate.

### 4.4. Determination of Intracellular ROS Generation

H9c2 cells were dispensed in 2 × 10^5^ cells/well within 6-well culture plates. Seabuckthorn flavonoids were added to H9c2 cell cultures at the 10 μM concentration for 1 h and then co-incubation was continued with 2.5 µM doxorubicin for 24 h in the logarithmic growth phase. After a series of the abovementioned treatment, cells were incubated in the dark under 37 °C for 30 min with 10 μM 2′,7′-dichlorofluorescein diacetate (DCFH-DA, Thermo Fisher Scientific Inc., Rockford, IL, USA), then washed twice with pre-warmed PBS, and analyzed using a NovoCyte 2040R flow cytometer (San Diego, CA, USA) with excitation wavelengths of 488 nm and 525 nm emission wavelengths [34]. 

### 4.5. ATP Content Measurement

Wells containing Dox were washed with PBS and refilled with DMEM prior to measuring ATP content [33]. ATP content was measured using a luciferase-based luminescence-enhanced ATP assay kit (Beyotime Biotech Inc., Shanghai, China). 

### 4.6. Transmission Electron Microscopy (TEM)

After a series of the abovementioned treatment, H9c2 cells were washed by PBS, and were digested by pancreatic enzymes, and centrifuged at 1000 rpm for 5 min; precipitates were collected and fixed with 2.5% glutaraldehyde at 4 °C for 2 h. After a series of ethanol dehydration steps, embedding, polymerization, sectioning, and staining were performed. Next, the 50–70 nm-thick sections were stained by uranyl acetate and citric acid, respectively (Leica EM UC7). Finally, images were taken and visualized by electron microscopy (HITACHI HT 7800 120kv, Tokyo, Japan) [35].

### 4.7. Western Blot Analysis

After a series of the abovementioned treatment, H9c2 cells were washed twice with ice-cold PBS, and were lysed in RIPA buffer with protease inhibitor PMSF on ice for 5 min, then were harvested to 1.5 mL centrifuge tubes. Lysis lasts for 25 min on ice. The homogenate was centrifuged at 12,000× *g* for 20 min at 4 °C. The supernatants were collected, and protein concentration was determined by the Pierce-23225 BCA protein assay kit. Equal amounts of proteins (30 μg from each sample) were separated by 10~15% SDS-PAGE gel electrophoresis and transferred to a polyvinylidene difluoride (PVDF) membrane. Membranes were blocked with 5% free-fat milk in TBS containing 0.05% Tween 20 (TBST) for 1 h at room temperature. Then, the membranes were washed for 10 min with TBST three times, incubated with primary antibodies (diluted 1:1000 in the blocking solution) overnight at 4 °C, and washed three times (10 min each) with TBST. Subsequently, the blots were incubated with horseradish peroxidase (HPR)-conjugated secondary antibodies for 1 h at room temperature. Membranes were washed as described above, incubated with ECL reagent, and visualized using a 5200 Multi Luminescent image analyzer (Tanon Science & Technology Co., Ltd., Shanghai, China). Primary antibodies for p-JNK, JNK, p-IRE1α, Mfn1, and Mfn2 were purchased from Abcam Inc. Primary antibodies for β-actin, p-Src, and Caspsae-3 were purchased from Cell Signaling Technology, Primary antibodies for Src and IRE1α were purchased from Affinity Biosciences. Primary antibodies for Sab was purchased from Novus Biologicals (NBP1-80796). HRP goat anti-rabbit Antibody and HRP goat anti-mouse antibody were purchased from Beijing BioDee Biotechnology Co., Ltd. (Beijing, China). 

### 4.8. Molecular Docking

Molecular docking was performed using AutoDock Vina (Docking number = 50, Scripps Research Institute, San Diego, CA, USA) to elucidate the binding mechanism between JNK1 (Protein Data Bank (PDB) ID: 4QTD) and three flavonoids from Hippophae rhamnoides. The 2D structure of the three flavonoids was drawn and converted to 3D for energy minimization by ChemDraw2019. The JNK1 target protein was prepared for molecular docking simulations by removing water molecules and co-crystallized ligands. The position of the original ligand in the X-ray structure of JNK1 was defined as the binding site, in which the energy of the ligand binding to JNK1 was used to evaluate the binding ability. PyMOL v.1.3 was used to visualize the results. All docked poses were ranked by affinity (kcal/mol) scoring function. The conformation with the lowest binding free energy was finally identified as the best probable binding mode. 

### 4.9. Statistics

All data were expressed as the mean ± SEM from three independent experiments. Statistical analysis was performed using one-way analysis of variance (ANOVA). 

## Figures and Tables

**Figure 1 ijms-24-03174-f001:**
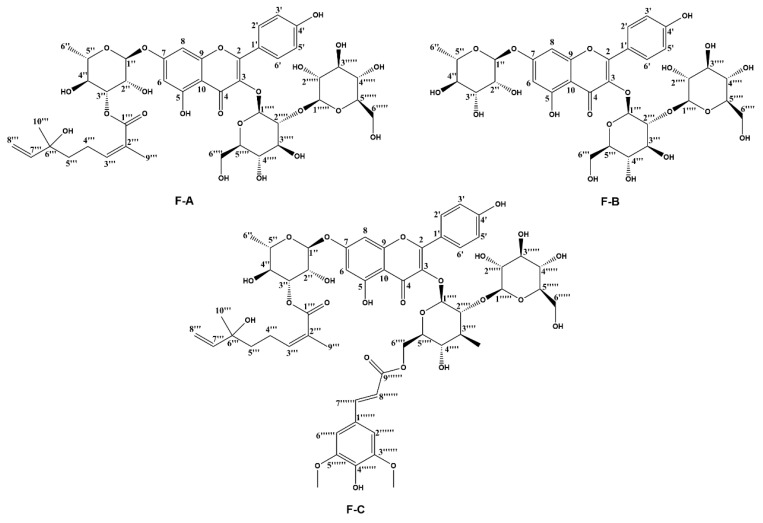
Chemical structures of three flavonoids (F-A, F-B, and F-C) from *Hippophae rhamnoides* L.

**Figure 2 ijms-24-03174-f002:**
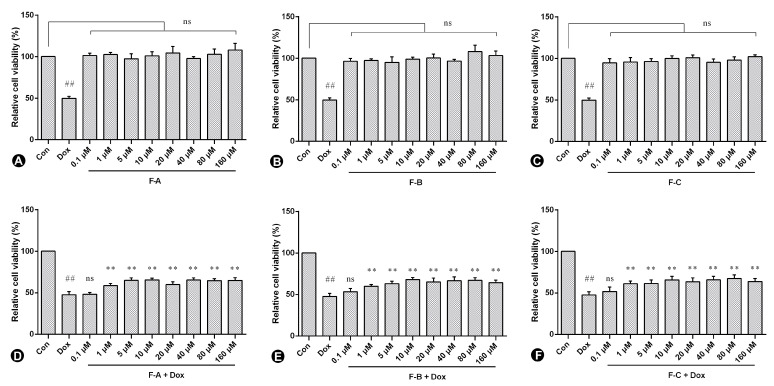
The effect of three flavonoids on Dox-induced cytotoxicity in H9c2 cells. Cell viability of H9c2 cells after exposure to different concentrations of F-A (**A**), F-B (**B**), and F-C (**C**) without Dox and the protective effect of F-A (**D**), F-B (**E**), and F-C (**F**) on Dox-induced cytotoxicity in H9c2 cells. Results are mean ± SEM from three independent experiments. ## *p* < 0.01 vs. control, ** *p* < 0.01 vs. Dox treated group. ns, not significant.

**Figure 3 ijms-24-03174-f003:**
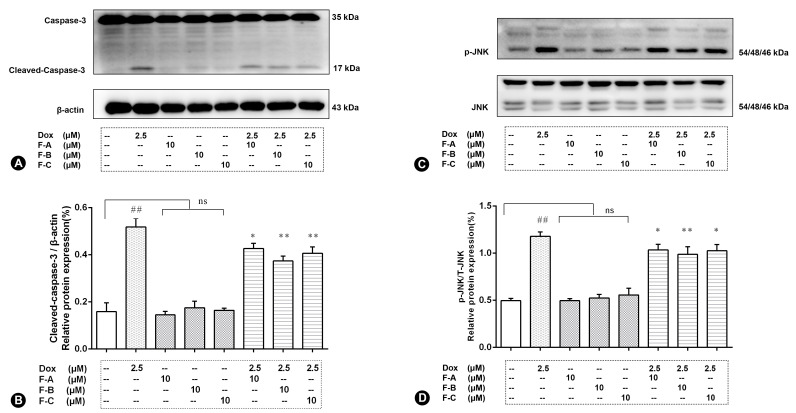
Flavonoids from *Hippophae rhamnoides* Linn affect the protein expression level of cleaved caspase-3 (**A**,**B**) and p-JNK (**C**,**D**) in Dox-induced H9c2 cells. Cells were pretreated with same concentrations of F-A, F-B, and F-C (10 µM) for 1 h, then exposed to Dox for 24 h. Relative folds of cleaved caspase-3, p-JNK. (**B**) The relative expression level of cleaved caspase-3 in Dox group significantly increased compared to the NC group (*p* < 0.01). There was no significant activation of cleaved caspase-3 in the F-A, F-B, and F-C group without Dox. But F-A, F-B, and F-C can significantly inhibit the Dox-induced activation of caspase-3 in comparison to the Dox model group (*p* < 0.01). (**D**) H9c2 cells treated with 2.5 μM Dox rapidly increased the relative expression level of phosphorylation of JNK (*p* < 0.01). Compared to the NC group, F-A, F-B, and F-C with the absence of Dox has no significant increase in the expression level of p-JNK. The expression levels of p-JNK compared to the Dox group was significantly suppressed by the administration of F-A, F-B, and F-C in the presence of 2.5 µM Dox (*p* < 0.01). Results are mean ± SEM from three independent experiments. ## *p* < 0.01 vs. control, * *p* < 0.05, ** *p* < 0.01 vs. Dox group. ns, not significant.

**Figure 4 ijms-24-03174-f004:**
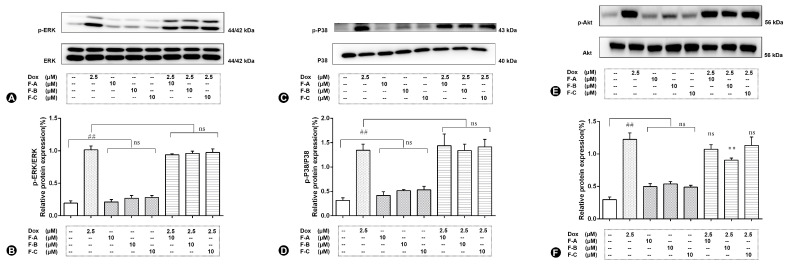
The effects of three flavonoids on p-ERK1/2 (**A**,**B**), p-P38 (**C**,**D**), and p-Akt (**E**,**F**) protein expression level of Dox-treated H9c2 cells. Cells were pretreated with same concentrations of F-A, F-B, and F-C (10 µM) for 1 h, then exposed to Dox for 24 h. Relative folds of p-Akt, p-ERK1/2, and p-P38. (**B,D,F**) The relative expression level of p-P38, p-ERK, and p-Akt in Dox group significantly increased compared to the NC group (*p* < 0.01). There was no significant activation of p-P38, p-ERK, and p-Akt in the F-A, F-B, and F-C group without Dox. However, F-A, F-B, and F-C had no significantly protective effect to reduce protein phosphorylation. Results are mean ± SEM from three independent experiments. ## *p* < 0.01 vs. control, ** *p* < 0.01 vs. Dox group. ns, not significant.

**Figure 5 ijms-24-03174-f005:**
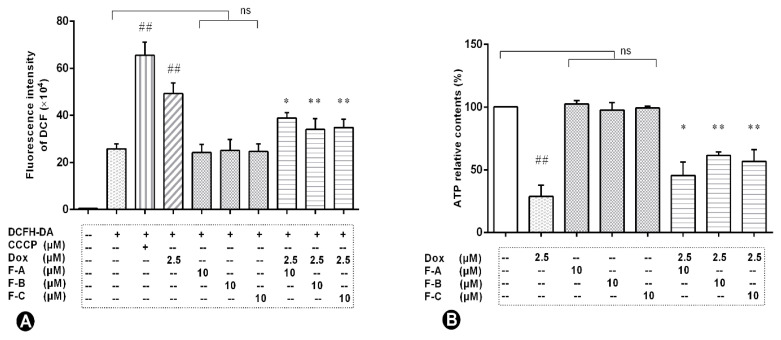
The effects of three flavonoids on intracellular ROS level (**A**) and ATP content (**B**) under Dox treatment in H9c2 cells. Cells were pretreated with same concentrations of F-A, F-B, and F-C (10 µM) for 1 h, then exposed to Dox for 24 h. CCCP (carbonyl cyanide 3-chlorophenylhydrazone) as a negative contrast group. (**A**) The DCF fluorescence intensity in Dox group significantly increased compared to the NC group (*p* < 0.01). There was no marked change of DCF fluorescence intensity in the F-A, F-B, and F-C group without Dox. But F-A, F-B, and F-C can significantly inhibit the Dox-induced elevation of ROS level in comparison to the Dox model group (*p* < 0.01). (**B**) H9c2 cells treated with 2.5 μM Dox rapidly reduced the content of ATP. Compared to the NC group, F-A, F-B, and F-C with the absence of Dox has no significant decrease (*p* > 0.05) the content of ATP. The content of ATP compared to the Dox group was significantly increased by the administration of F-A, F-B, and F-C in the presence of 2.5 µM Dox. Results are mean ± SEM from three independent experiments. ## *p* < 0.01 vs. control, * *p* < 0.05, ** *p* < 0.01 vs. Dox group. ns, not significant.

**Figure 6 ijms-24-03174-f006:**
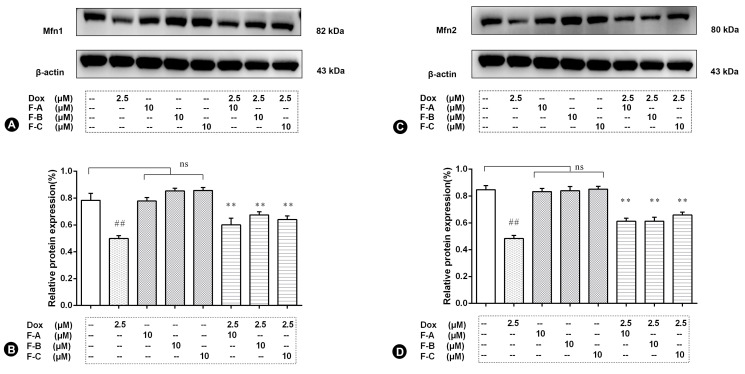
The effects of three flavonoids on mitochondrial fusion proteins Mfn-1 (**A**,**B**) and Mfn-2 (**C**,**D**) during Dox-induced cardiotoxicity. Cells were pretreated with same concentrations of F-A, F-B, and F-C (10 µM) for 1 h, then exposed to Dox for 24 h. Relative folds of Mfn1 (**C**) and Mfn2 (**D**). (**B**,**D**) The relative expression level of Mfn1and Mfn2 in Dox group significantly decreased compared to the NC group (*p* < 0.01). There was no significant change of Mfn1and Mfn2 in the F-A, F-B, and F-C group without Dox. But F-A, F-B, and F-C can significantly enhance the Dox-induced reduction of Mfn1 and Mfn2 in comparison to the Dox model group (*p* < 0.01). Results are mean ± SEM from three independent experiments. ## *p* < 0.01 vs. control, ** *p* < 0.01 vs. Dox group. ns, not significant.

**Figure 7 ijms-24-03174-f007:**
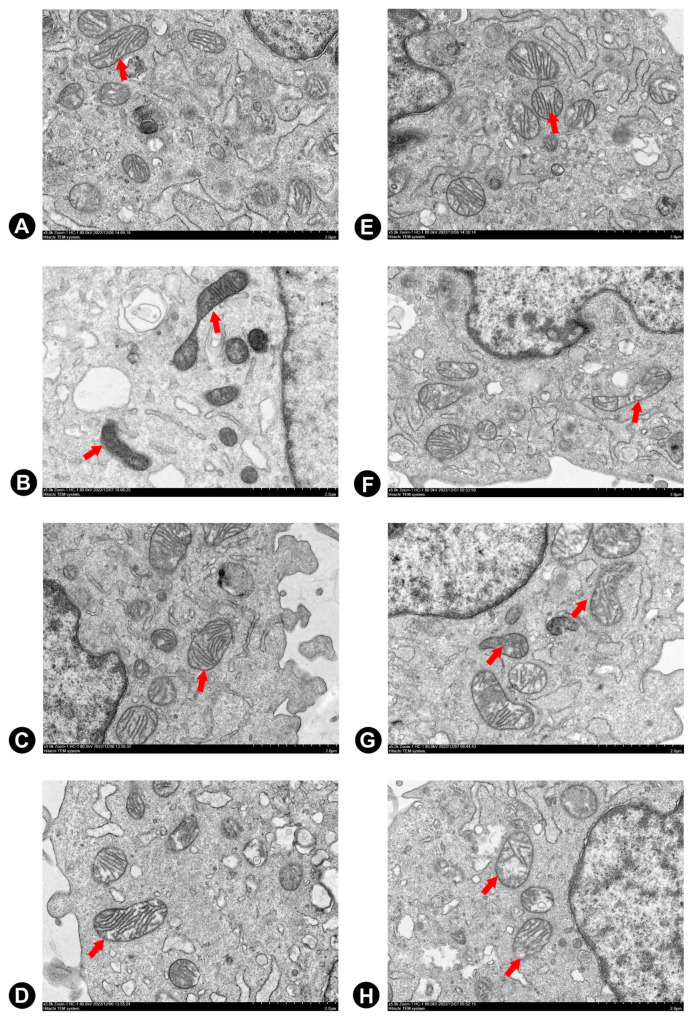
The effects of three flavonoids on mitochondrial ultrastructure during Dox-induced cardiotoxicity by SEM (5000×). Cells were pretreated with same concentrations of F-A, F-B, and F-C (10 µM) for 1 h, then exposed to Dox for 24 h. (**A**) NC Group; (**B**) 2.5 μM Dox Group; (**C**) F-A Group; (**D**) F-B Group; (**E**) F-C Group; (**F**) F-A + 2.5 μM Dox Group; (**G**) F-B + 2.5 μM Dox Group; and (**H**) F-B + 2.5 μM Dox Group.

**Figure 8 ijms-24-03174-f008:**
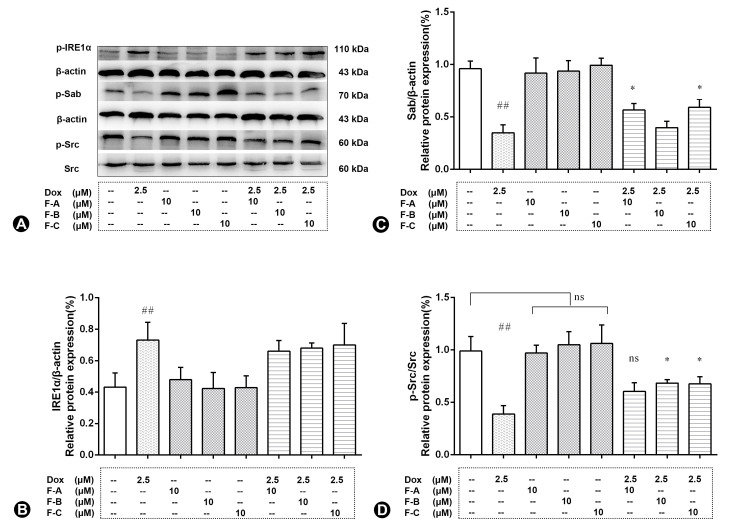
The effects of three flavonoids on Sab, p-Src, and p-IRE1α protein expression level of Dox-treated H9c2 cells. Cells were pretreated with same concentrations of F-A, F-B, and F-C (10 µM) for 1 h, then exposed to Dox for 24 h. Relative folds of Sab (**A**,**B**), p-Src (**A**,**D**), and p-IRE1α (**A**,**C**). (**B**) The relative expression level of p-IREα in Dox group significantly increased compared to the NC group (*p* < 0.01). There was no significant activation of p-IREα in the F-A, F-B, and F-C group without Dox. F-A, F-B, and F-C had no significantly protective effect to reduce Dox-induced p-IREα protein phosphorylation. (**C**) The relative expression level of Sab in Dox group significantly decreased compared to the NC group (*p* < 0.01). There was no significant change of Sab in the F-A, F-B, and F-C group without Dox. But F-A and F-C can significantly enhance the Dox-induced reduction of Sab in comparison to the Dox model group (*p* < 0.05). (**D**) The relative expression level of p-Src in Dox group significantly decreased compared to the NC group (*p* < 0.01). There was no significant change of p-Src in the F-A, F-B, and F-C group without Dox. But F-B and F-C can significantly enhance the Dox-induced reduction of p-Src in comparison to the Dox model group (*p* < 0.05). Results are mean ± SEM from three independent experiments. ## *p* < 0.01 vs. control, * *p* < 0.05 vs. Dox group. ns, not significant.

**Figure 9 ijms-24-03174-f009:**
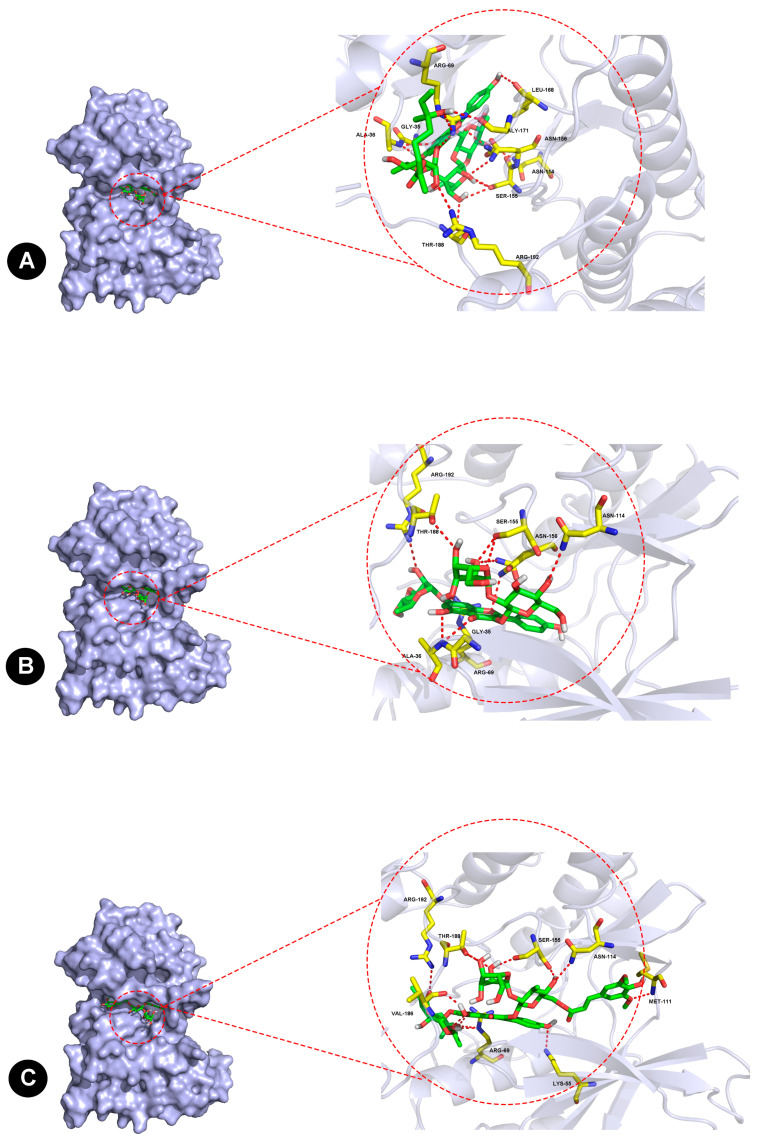
Molecular docking to predict the binding of three flavonoids to JNK-1 via AutoDock Vina, and PyMOL. Demonstration of the predicted binding conformation and corresponding interaction amino acid residues of F-A (**A**), F-B (**B**), and F-C (**C**) docking into JNK-1.

**Figure 10 ijms-24-03174-f010:**
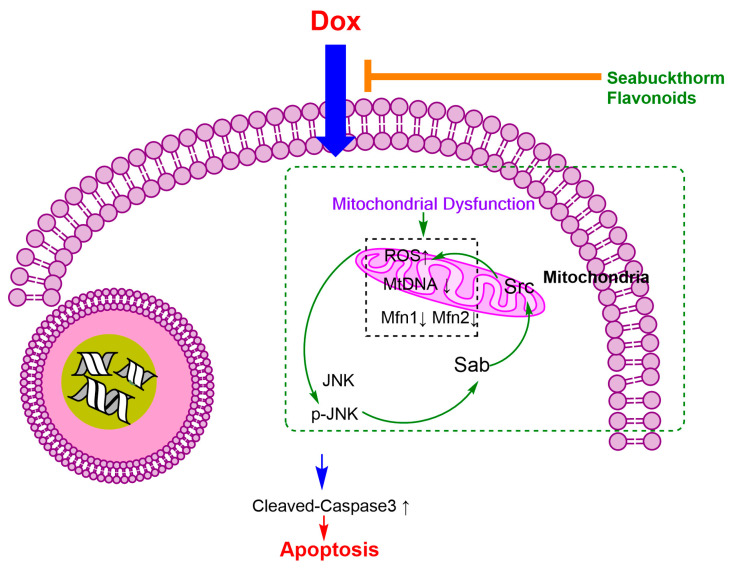
Possible signaling mechanism pathways.

## Data Availability

Not applicable.

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
