# Peer review of "Flavonoids from Hippophae rhamnoides Linn. Revert Doxorubicin-Induced Cardiotoxicity through Inhibition of Mitochondrial Dysfunction in H9c2 Cardiomyoblasts In Vitro"

_ijms, 2023, doi:10.3390/ijms24043174_

Round 1
Reviewer 1 Report
Dear Authors
The present experiment is a continuation of your other works in the field of the effect of ÙŽAlkaloids and Phenylparpenoid of this plant on the decreasing toxicity of doxorubicin on heart function. Despite designing and conducting the experiment with high accuracy and precision, there is some basic question as follows:
1) In theses three articles Phenylpropanoids (aromatic amino acid- Zhou et al., 2021) Alkaloids (Li et al., 2022) and Polyphenol (current expriment) of Hippophae rhamnoides, had exactly the same effect. How can the Alkaloids, Phenylpropanoids and Polyphenol of a plant have exactly the similar biological effect !?
Zhou, W., Ouyang, J., Hu, N., Li, G., & Wang, H. (2021). Protective effect of two alkaloids from Hippophae rhamnoides Linn. Against doxorubicin-induced toxicity in H9c2 Cardiomyoblasts. Molecules, 26(7), 1946.
Li, G., Chu, M., Tong, Y., Liang, Y., Wang, S., Ma, C., ... & Zhou, W. (2022). Protective Effects of Hippophae rhamnoides L. Phenylpropanoids on Doxorubicin-Induced Cardiotoxicity in Zebrafish. Molecules, 27(24), 8858.
2) If these polyphenols have strong antioxidant effect, why the effect of the concentration of 0.1 up to 100 (0.1, 10, 20, 40, 80, 160 umol) did not have any significant difference in all tests (Figure 2).
Other comments which is neccessary to be done are as below:
3) In the introduction, you should mention to these tow reference (Zhou et al., 2021; Li et al., 2022) and its results and highlight the novelty of the new work according to these works.
4) Ù‘Figure 2 shows that the effect of 0.1 and 10 micro mole of polyphenol on the cell toxicity of Dox was not significantly, why did you use 10 micro mole concentration in other tests?
Author Response
Thank you for your comments! The following is my answers and points.
- In theses three articles Phenylpropanoids (aromatic amino acid- Zhou et al., 2021) Alkaloids (Li et al., 2022) and Polyphenol (current expriment) of Hippophae rhamnoides, had exactly the same effect. How can the Alkaloids, Phenylpropanoids and Polyphenol of a plant have exactly the similar biological effect !?
- Zhou, W., Ouyang, J., Hu, N., Li, G., & Wang, H. (2021). Protective effect of two alkaloids from Hippophae rhamnoides Linn. Against doxorubicin-induced toxicity in H9c2 Cardiomyoblasts.Molecules, 26(7), 1946.
- Li, G., Chu, M., Tong, Y., Liang, Y., Wang, S., Ma, C., ... & Zhou, W. (2022). Protective Effects of Hippophae rhamnoides L. Phenylpropanoids on Doxorubicin-Induced Cardiotoxicity in Zebrafish. Molecules, 27(24), 8858.
Answer: Thank you for your question. Studies by other research groups have shown that flavonoids or alkaloids of Hippophae rhamnoides Linn. exhibit an excellent protective effect on injured cardiomyocytes in vitro or in vivo.
Abudureyimu Gulimire et al. reported that total flavonoids of Hippophae rhamnoides Linn. can increased myocardial GSH-Px and SOD activities and decreased MDA content to protect myocardial function against Dox-induced cardiotoxicity in rats [1]. In addition, total flavonoids of Seabuckthorn are used as pharmaceutical ingredients to treat cardiovascular diseases. Tongxinshu capsules mainly used in patients with coronary heart disease and angina pectoris are manufactured by Qinghai high tech Pharmaceu-tical Co., Ltd.
Changying Hu et al. and Deping Xu et.al. reported that total alkaloids from Hippophae rhamnoides Linn. exhibited an excellent protective effect on cultured neonatal rat cardiomyocytes injured by ischemia/reperfusion [2-3]. Hippophamide and N-p-coumaroyl-4-aminobutan-1-ol were obtained by purification and separation of seabuckthorn seed residue. Hippophamide has also been found by Yanmin Zhao in Bufonis Venenum [4]. Bufonis Venenum as an important Chinese traditional natural medicine, the chemical constituents were mainly included by bu fadienolides, indole alkaloids, steriods, etc. Clinical applications in the narcotic analgesic and cardiac therapy. Studies of Bufonis Venenum have also suggested hippophamide’s potential cardiogenic properties. The aboved results indicate that flavonoids and alkaloids of seabuckthorn have potential protective effects on Dox-induced cardiotoxicity.
Li Gang et.al. reported that phenylpropanoids from Hippophae rhamnoides L. can play a protective role in Doxorubicin-induced cardiotoxicity in zebrafish. The four phenylpropanoid monomers are p-coumaric acid (P1), chlorogenic acid (P2), caffeic acid (P3), and ferulic acid(P4) respectively. These four compounds are common compounds found in fruits, vegetables, grains, and fungi. They just happened to come from seabuckthorn in our previous studies and have universal antioxidant activity [5-8]. These four compounds also can be purified and isolated from most plants.
[1] Gulimire, A. et.al. Protective effect of total flavonoids of H. rhamnoides L. sunsp. turkestanica Rousi against adriamycin-induced cardiotoxicity in rats. J Xinjiang Med Univ 2010, 33(4), 383-385.
[2] Deping Xu et.al. Alkaloid against Ischemia/reperfusion Injury from Seeds of Hippophae rhamnoides Linn [J]. Nat Prod Res Dev, 2010, 22(6):937-939.
[3] Changying Hu et.al. Extraction, Isolation and Protective Effect of Alkaloid from Seabuckthorn Seeds on Injured Cardiomyocytes in Rats [J]. Food Science, 2010(9):4.
[4] Yanmin Zhao. Study on Chemical Constituents and Anti-tumor Activities of Bufonis Venenum [D]. Nanjing University of Chinese Medicine.
[5] Ying Liu et.al. Research progress of chlorogenic acid [J]. Journal of Chinese Medicinal Materials, 2012, 35(7):6.
[6] Wen Zhang, et.al. Advances in pharmacological effects of caffeic acid and its derivatives [J]. ProgressinVeterinaryMedicine, 2021. 42(8):103-106.
[7] Ke Liu. et.al. Research Progress of Phenolic Acid Compound Ferulic Acid [J]. Modern Chemical Research, 2022(20):5.
[8] Xiqin Guan, et.al. Research progress on pharmacological effects of p-coumaric acid [J]. Chinese Traditional and Herbal Drugs, 2018, 49(17):9.
(2) If these polyphenols have strong antioxidant effect, why the effect of the concentration of 0.1 up to 100 (0.1, 10, 20, 40, 80, 160 umol) did not have any significant difference in all tests (Figure 2). Other comments which is neccessary to be done are as below:
Answer: Toxic effects in H9c2 cells of separate treatment of seabuckthorn flavonoids (0.1~160μM) in the absence of 2.5 µM Dox were not significant difference compared with NC group (Fig2.A, Fig2.B, Fig2.C). Protective effect of F-A (Fig2.D), F-B (Fig2.E), and F-C (Fig2.F) on Dox-induced cytotoxicity in H9c2 cells were shown in 1~160μM concentration range.
(3) In the introduction, you should mention to these reference (Zhou et al., 2021; Li et al., 2022) and its results and highlight the novelty of the new work according to these works.
Answer: Thank you for your suggestions. The references mentioned above have been added in the article.
(4) Ù‘Figure 2 shows that the effect of 0.1 and 10 micro mole of polyphenol on the cell toxicity of Dox was not significantly, why did you use 10 micro mole concentration in other tests?
Answer: Toxic effects in H9c2 cells of separate treatment of seabuckthorn flavonoids (0.1~160μM) in the absence of 2.5 µM Dox were not significant difference compared with NC group. Protective effect of F-A (Fig2.D), F-B (Fig2.E), and F-C (Fig2.F) on Dox-induced cytotoxicity in H9c2 cells were shown in 1~160μM concentration range. On the one hand, we need to consider the range of clinically acceptable concentrations finally. On the other hand we get very small amounts of compounds, have to use carefully.

Reviewer 2 Report
The manuscript presented by Zhou et al (ijms-2151789) investigates the effects of three active compounds isolated from the Hippophae rhamnoides Linn extract (seabuckthorn seed) on cardiotoxicity associated with the use of doxorubicin in an in vitro model on H9c2 cells. The authors performed a number of studies in which they assessed the cytotoxicity of compounds, the effect on ROS generation, as well as a wide panel of Western Blot studies examining signaling pathways (e.g. JNK, MAPK). In addition, molecular docking of all three tested compounds to JNK1 was performed. The reviewer did not find any other reports in the literature on using these compounds in the study of DOX-induced cardiotoxicity. Thus, the topic undertaken by the authors is new and innovative. The references used by the authors are appropriate and the figures are legible. The manuscript has publication potential, however, the reviewer asks for correction or clarification of several reservations and issues.
1. Line 44 - "A possible molecular mechanism is widely accepted"
Is it the mechanism of action of doxorubicin or another compound? What molecular mechanism? The sentence is unclear to the reviewer and it is not clear what it refers to.
2. Line 57- "Seabuckthorn flavonoids extract"
The following phrase seems incorrect to the reviewer. It seems impossible to make a flavonoid extract from a plant. You can extract flavonoids from a plant extract or prepare a plant extract that contains many active ingredients, not only flavonoids.
3. Line 57-61 - Reference needed.
4. The introduction and abstract generally lack information about the studied plant (Seabuckthorn/Hippophae rhamnoides Linn.), where it occurs, etc. Moreover, there is no clear information that the above-mentioned two names of the plant can be used interchangeably.
5. Section 2.1.- the title is incomprehensible
6. Unfortunately, a reviewer would be skeptical to assume that these flavonoids abrogate the toxic effects of DOX on cardiomyocytes based on the MTT test. The graphs show that the cell viability of the combination of DOX and flavonoid is only slightly higher than that of DOX alone. There is statistical significance, however, the reviewer has doubts whether such a change is biologically noticeable.
7. In addition, the reviewer considers that the difference in band intensity between DOX and the combination of DOX and test compounds in the Western Blot in the apoptosis assay (cut caspase 3) is small. I do not know whether the statement that the tested compounds abolish the apoptosis caused by DOX is correct. The reviewer suggests densitometric measurement of the intensity of the bands and presenting the ratio to B-actin as a loading control.
8. Why were the cells treated with the compound only one hour prior to DOX administration? And not for a long time?
9. What was the reason for the authors' study of the JNK pathway? How did they come to the conclusion that this particular pathway may be related to the mechanism of DOX toxicity reversal?
10. Figure 8 should be placed under Section 2.5.
Author Response
Thank you for your comments! The following is my answers and points.
- Line 44 - "A possible molecular mechanism is widely accepted"
Is it the mechanism of action of doxorubicin or another compound? What molecular mechanism? The sentence is unclear to the reviewer and it is not clear what it refers to.
Answer: A possible molecular mechanism is widely accepted: doxorubicin can induce and promote reactive oxygen species (ROS) generation, in turn, lead to DNA damage or mitochondrial dysfunction. A colon ‘:’is used to introduce or explain what follows. The bold part is exactly what we want to express the possible mechanism.
- Line 57- "Seabuckthorn flavonoids extract"
The following phrase seems incorrect to the reviewer. It seems impossible to make a flavonoid extract from a plant. You can extract flavonoids from a plant extract or prepare a plant extract that contains many active ingredients, not only flavonoids.
Answer: Thank you for your suggestion. The description in the draft is not accurate. “Total flavonoids of seabuckthorn” replaces “Seabuckthorn flavonoids extract” in the revised manuscript.
- Line 57-61 - Reference needed.
Answer: The references have been added in the revised article.
- Shuaifeng Cheng, Xuming Yang. Clinical effect of Tibetan medicine Tongxinshu Capsule combined with Nicodil on angina pectoris of coronary heart disease [J]. Journal of Medicine and Pharmacy of Chinese Minorities, 2021 27(9):17-18.
- The introduction and abstract generally lack information about the studied plant (Seabuckthorn/Hippophae rhamnoides Linn.), where it occurs, etc. Moreover, there is no clear information that the above-mentioned two names of the plant can be used interchangeably.
Answer: Thank you for your suggestion. The purpose of this study was to focus on three flavonoids from Hippophae rhamnoides Linn. Plant information can be obtained from the paper of our previous study. This document has been quoted in the manuscript. ‘Seabuckthorn’ was used for ease of expression, after the fullname ‘Hippophae rhamnoides Linn. ‘was mentioned firstly.
- Zhou, W.; Yuan, Z.; Li, G.; Ouyang, J.; Suo, Y.; Wang, H., Isolation and structure determination of a new flavone glycoside from seed residues of seabuckthorn (Hippophae rhamnoides L.). Nat Prod Res 2018, 32, (8), 892-897.
- Section 2.1.- the title is incomprehensible
Answer: The title of section 2.1. was modified: “Flavonoids from Hippophae rhamnoides Linn. inhibits Dox-induced cytotoxicity in H9c2 Cells”.
- Unfortunately, a reviewer would be skeptical to assume that these flavonoids abrogate the toxic effects of DOX on cardiomyocytes based on the MTT test. The graphs show that the cell viability of the combination of DOX and flavonoid is only slightly higher than that of DOX alone. There is statistical significance, however, the reviewer has doubts whether such a change is biologically noticeable.
Answer: 2.5 µM Dox concentration can significantly damage H9c2 cells. This damage repair does not return to normal levels. Compared to the Dox model group, cell viability of the combination of Dox and flavonoids all both significantly increased (P<0.01).
- In addition, the reviewer considers that the difference in band intensity between DOX and the combination of DOX and test compounds in the Western Blot in the apoptosis assay (cut caspase 3) is small. I do not know whether the statement that the tested compounds abolish the apoptosis caused by DOX is correct. The reviewer suggests densitometric measurement of the intensity of the bands and presenting the ratio to β-actin as a loading control.
Answer: This Cleaved Caspase-3 antibody detects endogenous levels of the large fragment (17 kDa) of activated caspase-3, also detects full-length caspase-3 (CST #14220). Compared to the sensitivity cleaved caspase-3 Rabbit mAb, a slightly differences in cleaved caspase-3 in background can mean a big change.
The following figures of other cell line by our research group were detected by same cleaved caspase-3 antibodies
[1] Han Q, Han L, Tie F, et al. (20S)-Protopanaxadiol Ginsenosides Induced Cytotoxicity via Blockade of Autophagic Flux in HGC-27 Cells[J]. Chemistry & Biodiversity, 2020, 17(7).
- Why were the cells treated with the compound only one hour prior to DOX administration? And not for a long time?
Answer: Flavonoids from Hippophae rhamnoides Linn. were added to H9c2 cell cultures at the desired concentration 1 h before treatment with doxorubicin and then incubation was continued with 2.5 µM doxorubicin for 24 h. The procedure is adjusted according to the following references.
[1] Guo R , Wu K , Chen J , et al. Exogenous hydrogen sulfide protects against doxorubicin-induced inflammation and cytotoxicity by inhibiting p38MAPK/NFκB pathway in H9c2 cardiac cells.[J]. Cellular Physiology and Biochemistry, 2013, 32(6):1668-1680.
[2] Priya L B , Baskaran R , Huang C Y , et al. Neferine ameliorates cardiomyoblast apoptosis induced by doxorubicin: possible role in modulating NADPH oxidase/ROS-mediated NFκB redox signaling cascade[J]. Rep, 2017, 7(1).
[3] Kim D S , Woo E R , Chae S W , et al. Plantainoside D protects adriamycin-induced apoptosis in H9c2 cardiac muscle cells via the inhibition of ROS generation and NF-kappaB activation.[J]. Life Sciences, 2007, 80(4):314-323.
- What was the reason for the authors' study of the JNK pathway? How did they come to the conclusion that this particular pathway may be related to the mechanism of DOX toxicity reversal?
Answer: The semiquinone form of Dox is considered as a toxic short-lived metabolite, which can interact with mitochondrial enzymes urge to accumulate a lot of ROS. The results of MTT cell viability, ROS level and ATP content were suggested that flavonoids from Hippophae rhamnoides Linn. can ameliorate Dox-induced H9c2 cell damage. We hypothesized that this protective effect may be related to MAPK pathway,a large number of references indicate that MAPK signaling pathway plays a key role. Then, we further explored and investigated the protein expression levels of p-JNK, p-ERK, p-P38, p-Akt. F-A, F-B and F-C treated H9c2 cells significantly inhibited the Dox-induced activations of JNK. However, there is no significant reduced the in the levels of p-ERK1/2 and p-P38 in comparison with Dox group. We speculate that three flavonoids reduce Dox-induced apoptosis may be achieve through down regulation of ROS-dependent JNK activation that further promotes cell survival, rather than down regulation of ERK, P38 and p-Akt activation. Simultaneously, the results showed that flavonoids from Hippophae rhamnoides Linn. can alleviate Dox-induced mitochondrial structure and function disorders. Sab, a c-Jun N-terminal kinase (JNK)-interacting protein, locates at mitochondrial outer membrane, which mediates mutual amplifcation of mitochondrial ROS generation and JNK activation. That's what reason we focused on the mitochondrial JNK-Sab-ROS positive feedback loop to elucidate the mechanism of Dox-induced apoptosis.
[1] Brantley-Finley C , Lyle C S , Du L , et al. The JNK, ERK and p53 pathways play distinct roles in apoptosis mediated by the antitumor agents vinblastine, doxorubicin, and etoposide[J]. Biochemical Pharmacology, 2003, 66(3):459-469.
[2]Hirata, Y., Inoue, A., Suzuki, S., Takahashi, M., Matsuzawa, A., Matsui, R., et al. Trans-Fatty acids facilitate DNA damage-induced apoptosis through the mitochondrial JNK-Sab-ROS positive feedback loop. Sci Rep 2020, 10, 2743.
- Figure 8 should be placed under Section 2.5.
Answer: Adjusted the order in which graphs appear in the text.

Round 2
Reviewer 1 Report
The authors have applied all the referee's comments in the revised article and now, the article has the necessary quality to be published in the valuable journal of IJMS.
Author Response
Thank you for your comments!
Reviewer 2 Report
The reviewer accepts the manuscript after these major revisions.
Author Response
Thank you for your comments!